# Intervention Effect of Aerobic Exercise on Physical Fitness, Emotional State and Mental Health of Drug Addicts: A Systematic Review and Meta-Analysis

**DOI:** 10.3390/ijerph20032272

**Published:** 2023-01-27

**Authors:** Xing Ye, Renyi Liu

**Affiliations:** School of Physical Education, China University of Geosciences (Wuhan), Wuhan 430074, China

**Keywords:** aerobic exercise, physical fitness, emotional state, mental health, drug addicts, meta-analysis

## Abstract

This study aims to discuss evidence for the efficacy of aerobic exercise in reducing drug addiction and improving the physical and mental health of drug addicts. We systematically searched several online databases as of the end of September 2022, including PubMed, EMBASE, the Cochrane Library, Web of Science, Scopus, Science Direct, CNKI, VIP, CBM, and Wanfang. All articles were identified, screened and included according to the inclusion or exclusion criteria. The Cochrane Handbook for Systematic Reviews of Interventions was used as a criterion for assessing the methodological quality of included studies. Random and fixed effects models were used for the analysis of standard mean differences (SMD) or mean differences (MD) with their 95% confidence intervals (CI). A total of 27 studies involving 2022 drug addicts were finally included in the analysis. The results of the meta-analysis showed that aerobic exercise could improve the physical fitness [body fat percentage: MD = −0.74, 95%CI (−1.40, −0.08), vital capacity: MD = 213.79, 95%CI (46.28, 381.29), muscle force: MD = 1.21, 95%CI (0.34, 2.08), flexibility: MD = 4.61, 95%CI (2.98, 6.25), balance: MD = 9.95, 95%CI (6.29, 13.62)], regulate the systolic blood pressure: MD = −4.38, 95%CI (−7.08, −1.68), diastolic blood pressure: MD = −2.66, 95% CI (−3.82, −1.51), beats per minute: MD =−1.92, 95%CI (−3.19, −0.65); emotional state [anxiety: MD = −4.56, 95% CI (−5.67, −3.45), depression: MD = −3.28, 95%CI (−5.16, −1.39), drug craving: SMD= −1.68,95% CI(−2.56, −0.80)], and promote the mental health [anxiety: MD = −0.22, 95%CI (−0.33, −0.11), obsessive−compulsive: MD = −0.26, 95%CI (−0.50, −0.03), somatization: MD = −0.21, 95%CI (−0.27, −0.14), depression: MD = −0.21, 95%CI (−0.28, −0.15), psychoticcism: MD = −0.12, 95%CI (−0.18, −0.06), phobic anxiety: MD = −0.11, 95%CI (−0.16, −0.07), paranoid ideation: MD = −0.09, 95%CI (−0.15, −0.02), interpersonal sensitivity: MD = −0.16, 95%CI (−0.22, −0.10), hostility: MD = −0.12, 95%CI (−0.18, −0.05)], with statistically significant differences(*p* < 0.05)] of drug addicts. Thus, aerobic exercise could effectively improve the physical fitness, emotional state and mental health of drug addicts, and reduce their drug addiction. For clinical practitioners and researchers, this study could provide more reliable evidence for addiction treatment.

## 1. Introduction

One of the most serious problems in the world is drug addiction, which has been classified as a world public health hazard. Drugs refer to heroin, methamphetamine, marijuana, cocaine and other illegal substances that can cause addiction under the control of the state. Drugs can trigger abnormal excitatory or inhibitory effects in the human central nervous system, with a range of neurological and psychiatric symptoms [1,2]. They may disrupt the delicate balance that exists between the energy-producing metabolic system and the inherent cytoprotective mechanisms, putting drug addicts at high risk for the development of metabolic syndrome when taking drugs. Drug addiction is closely related to the onset and recurrence of chronic mental disorders and the individual’s physical, psychological and environmental conditions. It is also easy to increase the risk of accidental injury, suicide and death due to excessive use [3]. According to the United Nations Office on Drugs and Crime World Drug Report 2022, about 300 million people worldwide used drugs in this year [4]. More than 36 million people were living with drug abuse disorders. Between 2010 and 2022, the number of drug addicts increased by 22%. At the same time, sharing syringes increases the transmission of multiple viruses among drug users, such as HIV and hepatitis B virus [5,6]. Thus, drug addiction has become a global problem that endangers personal health and family stability, threatens public health and affects social order. Currently, the main treatment methods for drug addicts include pharmacotherapy, cognitive behavioral therapy and psychological intervention methods. Although these methods are commonly available in drug abuse treatment programs, there is pessimism in this area about the efficacy of these treatments for drug use disorders. This frustrating situation prompts the reconsideration of therapeutic strategies against drug addicts.

Non-pharmacological intervention has become a priority in the treatment of drug addicts, and exercise interventions have been suggested as an important measure of physical therapies. The American College of Sports Medicine has repeatedly and strongly recommended exercise interventions as an effective way to promote health and treat disease. Animal studies and human trials have shown that exercise can effectively prevent addiction, inhibit drug-seeking behaviors, help drug addiction and prevent relapse [7,8]. Among all types of exercises associated with the treatment of drug addiction, aerobic exercise could be the first choice, with the advantages of low cost and few side effects, and being the most commonly used. In particular, it has demonstrated some inspiring results in the treatment practice of drug addicts. According to previous studies, regular aerobic exercise could regulate a variety of psychological symptoms in drug addicts, such as anxiety, depression, paranoia, hostility or compulsiveness, and bring many benefits in physiological and psychological health. Aerobic exercise can improve physical fitness, reduce withdrawal symptoms, increase the quality of life, decrease daily usage and craving, and enhance the therapeutic effect of detoxification [9]. A recent meta-analysis suggests that traditional Chinese health-promoting exercise is effective in improving physical and mental health in drug user disorders [10]. However, there are some limitations, such as the intervention mode of the included research being globally unpopular, and there is a lack of publication bias tests. The current literature on aerobic exercise interventions for drug addiction has reported fewer large-sample, multicenter rigorous randomized controlled trials, and further organized studies are still needed [11]. Based on the lack of convincing analysis and evaluation, it is necessary to seek more evidence related to the intervention effect of aerobic exercise. In view of this condition, the purpose of this study was to explore the effects of aerobic exercise on the physical fitness, emotional state, and mental health of drug addicts through a systematic review and meta-analysis.

## 2. Methods

This study was conducted according to the Preferred Reporting Items for Systematic Reviews and Meta-Analyses (PRISMA) statement and the Cochrane Handbook of Systematic Reviews.

### 2.1. Literature Search Strategy

A comprehensive search was conducted as of the end of September 2022, through the following databases: PubMed/MEDLINE, Scopus, EMBASE, Science Direct, Cochrane Library, Web of Science, China National Knowledge Infrastructure (CNKI), Chinese Biomedical Literature Database (CBM), WanFang, Weipu Database (VIP). Randomized controlled trials (RCTs) of aerobic exercise interventions for drug addicts were identified. Any research that may qualify was considered for review regardless of the publishing language, population, or main results. The databases were searched using a combination of subject terms and keywords. The literature was identified after repeated pre-screening, supplemented by hand searching and tracing references of included literature when necessary. Database searches were limited to studies assessing humans and adults, and conducted using the following keywords: aerobic exercise, physical training, drugs, detoxification, heroin, cannabis, methamphetamine, rehabilitation, drug addiction, randomized controlled traits.

### 2.2. Study Selection

To be included in the systematic review and meta-analysis, studies had to fulfill the following inclusion criteria. Inclusion criteria included study participants aged 18 years or above who were drug dependent and met the diagnostic criteria for illicit drug abusers; the intervention duration and the sample size were unlimited, and all the participants were involved in aerobic exercise. The outcome indicators of this study involved physical fitness (body composition (body mass index, BMI; mass; body fat percentage), cardiorespiratory function (systolic blood pressure, diastolic blood pressure, beats per minute, vital capacity), muscle force (grip strength), flexibility (sit-and-reach), balance (one-leg stand with eye closed)), emotional state (anxiety (self-rating anxiety scale, SAS), depression (self-rating depression scale, SDS), drug craving (brief cocaine craving questionnaire, CCQ; desire for speed questionnaire, DSQ; visual analog scale, VAS)), mental health (symptom checklist-90, SCL-90). Exclusion criteria included duplicate published studies, studies with unqualified results, and those without appropriate outcome indicators.

### 2.3. Data Extraction

Two reviewers independently screened the title and abstract of the studies and then performed full-text reading and study selection based on eligibility criteria. The extracts included basic information (first author, year of publication, sample size, age), intervention aspects (treatment duration, intensity, frequency, time, comparison details), and outcome indicators. Disagreements between the two reviewers were resolved in discussions with the senior author. Authors of original publications were contacted for additional information if necessary.

### 2.4. Methodological Quality Assessment

The methodological quality of the included studies was assessed by using the Cochrane risk of bias tool, with assessment form entries containing random sequence generation, allocation concealment, blinding of participants and personnel, blinding of outcome assessment, incomplete outcome data, selective reporting, and other biases.

### 2.5. Statistical Analysis

All the data extracted from each study were analyzed by Stata 16.0 software and Review Manager 5.4 software. The heterogeneity of included studies was assessed using the I^2^ statistic. Fixed-effect (I^2^ <50% or *p* value > 0.1) and random-effect models (I^2^ > 50% or *p* value < 0.1) were used as analysis models based on the heterogeneity results. Effect values were counted using the standardized mean of difference (SMD), mean difference (MD) and 95% confidence interval (CI). In addition, a *p* value less than 0.05 in the Egger test for publication bias indicated publication bias sensitivity analysis tests for the stability of the results by excluding each study consecutively.

## 3. Results

### 3.1. Literature Search

According to the search strategy mentioned above, 1568 records were retrieved from the databases. After deleting duplicate records, there were 956 records left. The titles and abstracts were reviewed, and 863 studies that did not meet the standards were excluded. We reviewed the full text of the remaining 93 studies, and further excluded 66 studies. Finally, we included 27 studies for systematic review and meta-analysis (Figure 1).

### 3.2. Characteristics of Studies

Twenty-seven RCTs were included in this review, involving 2022 drug addicts. The sample size included in the studies ranged from 22 to 200, and the duration of aerobic exercise intervention ranged from 2 weeks to 32 weeks, with a frequency of 3~7 and 15~120 min each week. The basic information of the included studies is shown in Table 1.

### 3.3. Methodological Quality

We evaluated the bias risk of all included studies according to the Cochrane collaboration tool. The risk of bias in the included studies is shown in Figure 2.

### 3.4. Meta-Analysis Results

#### 3.4.1. Physical Fitness 

##### Body Compositions

In this meta-analysis, there were eighteen studies (a total of 1075 drug addicts) involving aerobic exercise interventions for body composition in drug users, with the main indicators including BMI, body fat percentage and mass. The results of heterogeneity analysis suggested that BMI studies should use a random effects model (I^2^ = 80%, *p* < 0.00001) and studies on body fat percentage (I^2^ = 1%, *p* = 0.43) and mass (I^2^ = 0%, *p* = 0.44) should use a fixed effects model. The results of the meta-analysis showed that there was a statistical significance between aerobic exercise and the control group in terms of body fat percentage in drug addicts [MD = −0.74, 95%CI (−1.40, −0.08), *p* = 0.03] (Figure 3), and no significant difference in BMI [MD = −0.74, 95%CI (−1.54, 0.06), *p* = 0.07], and mass [MD = −1.32, 95%CI (−2.87, 0.22), *p* = 0.09], as shown in Figure 4 and Figure 5.

##### Cardiopulmonary Function

Nineteen studies (a total of 1141 drug addicts) reported the results of cardiopulmonary function evaluation, including vital capacity, systolic blood pressure, diastolic blood pressure and beats per minute. The results of heterogeneity analysis showed that the studies of vital capacity (I^2^ = 80%, *p* < 0.00001) and systolic blood pressure (I^2^ = 61%, *p* = 0.003) should use a random effect model; and the studies of diastolic blood pressure (I^2^ = 17%, *p* = 0.28) and beats per minute (I^2^ = 0%, *p* = 0.66) should use a fixed effects model. The pooled effect of these studies showed that in vital capacity [MD = 213.79, 95%CI (46.28, 381.29), *p* = 0.01], systolic blood pressure [MD = −4.38, 95%CI (−7.08, −1.68), *p* = 0.001], diastolic blood pressure [MD = −2.66, 95%CI (−3.82, −1.51), *p* < 0.00001], and beats per minute [MD= −1.92, 95%CI (−3.19, −0.65), *p* = 0.003], the aerobic exercise intervention groups had better rehabilitation results than the control groups, as shown in Figure 6, Figure 7, Figure 8 and Figure 9.

##### Muscle Force

Eleven studies (a total of 644 drug addicts) measured the effects of aerobic exercise on grip strength. The results of heterogeneity analysis suggested that grip strength studies should use a fixed effects model (I^2^ = 31%, *p* = 0.15). The pooled effect size showed significant effects of aerobic exercise intervention on muscle force [MD = 1.21, 95% CI (0.34, 2.08), *p* = 0.007], as shown in Figure 10.

##### Flexibility

Twelve studies (a total of 752 drug addicts) measured the effects of aerobic exercise on flexibility for sit-and-reach. The results of the heterogeneity analysis suggested that sit-and-reach studies should use a random effects model (I^2^ = 58%, *p* = 0.006). The pooled effect size showed significant effects of aerobic exercise on flexibility [MD = 4.61, 95%CI (2.98, 6.25), *p* < 0.00001], as shown in Figure 11.

##### Balance

Sixteen studies (a total of 1010 drug addicts) used a one-leg stand with eyes closed as an index to evaluate balance ability. The results of the heterogeneity analysis suggested that one-leg stand with eye closed studies used a random effects model (I^2^ = 83%, *p* < 0.00001). The results showed that after the aerobic exercise intervention, the experimental group had a longer standing time on one leg with eyes closed than the control group [MD = 9.95, 95%CI (6.29, 13.62), *p* < 0.00001], as shown in Figure 12.

#### 3.4.2. Emotional State

##### Anxiety

Ten studies (a total of 875 drug addicts) reported the effect of aerobic exercise on the SAS of drug addicts, and the combined analysis adopted a fixed effects model (I^2^ = 19%, *p* = 0.26). The results show that aerobic exercise interventions could significantly reduce the anxiety of drug addicts [MD = −4.56, 95%CI (−5.67, −3.45), *p* < 0.00001], as shown in Figure 13.

##### Depression

Ten studies (a total of 819 drug addicts) reported the effect of aerobic exercise on SDS of drug addicts, and the combined analysis adopted a random effect model (I^2^ = 72%, *p* = 0.0002). The results showed that aerobic exercise interventions could effectively alleviate the depression of drug addicts [MD = −3.28, 95%CI (−5.16, −1.39), *p* = 0.0007], as shown in Figure 14.

##### Drug Craving

Nine studies (a total of 554 drug addicts) used three scales as indicators to evaluate drug cravings. The pooled analysis using a random effects model (I^2^ = 95%; *p* < 0.00001) showed evidence of significant differences between the exercise group [SMD = −1.68 (−2.56, −0.80), *p* = 0.0002] and the control group, as shown in Figure 15.

#### 3.4.3. Mental Health

The SCL-90 study included nine items and explored the therapeutic effects of aerobic exercise on the mental health of drug addicts (a total of 558 participants). In SCL-90, nine indicators of mental health symptoms of drug users were included, including anxiety, psychoticism, phobic anxiety, paranoid ideation, obsessive-compulsiveness, somatization, interpersonal sensitivity, depression, and hostility. The results of heterogeneity analysis showed that the studies of anxiety (I^2^ = 63%, *p* = 0.008) and obsessive-compulsiveness (I^2^ = 94%, *p* < 0.00001) should use a random effect model; the studies of somatization (I^2^ = 38%, *p* = 0.13), depression (I^2^ = 23%, *p* = 0.24), psychoticism (I^2^ = 0%, *p* = 0.62), phobic anxiety(I^2^ = 0%, *p* = 0.52), paranoid ideation (I^2^ = 0%, *p* = 0.43), interpersonal sensitivity (I^2^ = 0%, *p* = 0.50), and hostility (I^2^ = 0%, *p* = 0.64) should use a fixed effects model. Meta-analysis showed that the therapeutic effect of aerobic exercise on mental health (SCL-90) of drug addicts was significantly superior to that of the control group. The specific manifestations were as follows: anxiety [MD = −0.22, 95%CI (−0.33, −0.11), *p* < 0.0001], obsessive-compulsiveness [MD= −0.26, 95%CI (−0.50, −0.03, *p* = 0.03)], somatization [MD = −0.21, 95%CI (−0.27, −0.14), *p* < 0.00001], depression [MD = −0.21, 95%CI (−0.28, −0.15), *p* < 0.00001], psychoticism [MD = −0.12, 95%CI (−0.18, −0.06), *p* < 0.0001], phobic anxiety [MD= −0.11, 95%CI (−0.16, −0.07), *p* < 0.00001], paranoid ideation [MD = −0.09, 95%CI (−0.15, −0.02), *p* = 0.007], interpersonal sensitivity [MD = −0.16, 95%CI (−0.22, −0.10), *p* < 0.00001], and hostility [MD = −0.12, 95%CI (−0.18, −0.05), *p* = 0.0007], as shown in Figure 16.

### 3.5. Sensitivity Analysis

Sensitivity analysis was carried out by excluding each of the 27 included studies, and the results did not show essential changes. As such, we considered the results in this study reliable and stable.

### 3.6. Publication Bias

In this study, the Egger test was used to analyze the vital capacity index, SAS index of emotional state, and interpersonal index of mental health of drug addicts during aerobic exercise interventions. Compared with the funnel plot method, the Egger test overcomes the drawbacks of subjective judgments of funnel plots and the results were more objective. The *p*-values for vital capacity (*p* = 0.716), SAS (*p* = 0.760), and interpersonal relationships (*p* = 0.096) were tested to be greater than 0.05, indicating that there was no publication bias.

## 4. Discussion

This meta-analysis was conducted to estimate the effects of an aerobic exercise intervention on the physical fitness, emotional state and mental health of drug addicts, and provide an overview of the latest evidence through 27 randomized controlled studies. Our pooled analysis showed that aerobic exercise had a significant beneficial effect on the emotional state and psychological well-being of drug addicts, with improvements in aspects of the evaluation indexes of physical fitness.

Firstly, our study showed that aerobic exercise was significantly better than the control group in enhancing cardiorespiratory fitness, muscle force and balance and flexibility in drug addicts. Previous studies have shown a significant cardioprotective effect of aerobic exercise. Aerobic exercise couldimprove oxygen utilization and blood supply to the heart, increases lung vital capacity, and contributes to cardiopulmonary function and physical fitness in drug addicts [39]. Wei [40] found that aerobic exercise produced significant changes in the ability of seated forward bending, reaction time, grip strength, and one-legged standing time in the rehabilitators, as well as positive effects on the morphology and physiological functions of the drug addicts after Chinese traditional exercise, i.e., the Yijinjing exercise intervention for drug rehabilitators in a women’s compulsory segregation facility. Jiang’s [41] research had shown that drug addicts undergoing aerobic exercise therapy had improved fitness, lowered blood pressure, heart rate, and improved flexibility while increasing aerobic endurance levels. Analysis of different outcome indicators showed significant improvements in cardiopulmonary function, such as vital capacity, diastolic blood pressure, and systolic blood pressure. This indicates that regular stimulation of appropriate intensity leads to an increase in the adaptability of the individual’s cardiopulmonary system function, an increase in cardiac output per beat, and an improvement in cardiovascular elasticity, resulting in an overall enhancement effect [42]. In this study, the improvement in muscle strength after aerobic exercise intervention in drug addicts was also quite significant, which could be related to the increase in muscle weight and decrease in body fat content, and on the other hand to the optimization of muscle fiber type and volume [43]. The improvement of balance function and flexibility could be related to appropriate joint extrusion and body control, thereby enhancing the proprioception input of drug addicts. Meanwhile, the warm-up and recovery before and after the exercise intervention encourage them to actively stretch muscles and shift the body’s center of gravity. The results were not statistically significant for body composition indicators regarding mass and BMI, and the advantages of aerobic exercise in these aspects could not be determined. This might be attributed to different exercise protocols, which were insufficient to produce significant improvements in body composition indicators. Therefore, more well-designed, high-quality and large-sample RCTs are needed to confirm these findings.

Next, the results of this study showed that aerobic exercise was effective in alleviating anxiety and depressive symptoms and reducing the craving for drugs in drug-addicted individuals. Drug craving is a core symptom of drug addiction and an important indicator of clinical diagnosis. Anxiety and depression drive addicts to become more dependent on drugs and are the most common negative emotions associated with drug addiction. The phenomenon of depression and anxiety in drug addicts concerns ambivalence [44]. They clearly understand that drug use will erode their health and harm society. Still, the physical and mental discomfort brought by drug addiction stimulates their thirst for drugs, and there is nowhere to vent this ambivalence, which eventually leads to anxiety and depressive tendencies in drug addicts. Studies have shown that aerobic exercise raises body temperature, which induces the release of endorphins, which are effective in reducing anxiety, depression and craving in drug addicts [45]. The findings of this study are consistent with the analysis by Wang et al. [46]. The intervention mechanism could be that aerobic exercise caused changes in adrenaline, and reduced 5-HT neurons induced by the serotonin reuptake inhibitors, thus promoting dopamine release and alleviating adverse emotions such as anxiety, depression and craving [47]. The results of this analysis confirm previous related studies and strengthen the reliability of the present findings, which are of great value for the application of aerobic exercise in clinical treatment.

Finally, the results of this study showed that aerobic exercise was effective in promoting significant improvements in the mental health of drug addicts, and the differences in SCL-90 scores were statistically significant. Drug addicts may be emotionally depressed due to their families, living environment, etc., and their mental health outlook may not be optimistic. In many studies on the mental health of drug addicts, the SCL-90 has become an important research tool that can effectively screen for common psychological problems through investigations, usually adopting standardized measurement tools such as the SCL-90, and taking samples by random or convenient sampling to carry out the test. Although this avoids the subjectivity of the results, it is difficult to guarantee the representativeness of the samples, which leads to the poor ecological validity of the research and the difficulty in popularizing the conclusions [48]. Given the limitations of previous studies, a quantitative analysis that is representative and leads to broad and general conclusions is particularly necessary. It is necessary to improve mental health, inducing peak experiences or higher states of consciousness to replace the appeal of drug-induced pleasure, improving self-efficacy by preventing destructive or maladaptive behaviors before they occur, and building better self-esteem relationships and mutual understanding between the addicted individual and society. Chu et al. [49] also indicated that exercise could positively affect the emotional state of drug addicts, weakening their feelings of shame and inferiority, remodeling correct ideals and beliefs, improving the sense of well-being, and contributing to improved mental health. Thus, it can be inferred that, because it can help individuals establish a better self-esteem relationship with society, aerobic exercise intervention is beneficial to the significant improvement of somatization, phobic anxiety, obsessive-compulsive symptoms, psychoticism, paranoia, interpersonal relationship, hostility, depression, and anxiety.

## 5. Limitations

There are limitations to our study. Due to the limitations of existing studies, only published RCTs from the libraries to date were included, and unpublished papers were not obtained and included; the lack of clear study information in the interpretation of some of the included literature affects the quality of the studies and reduces the validity of the assessment of clinically relevant data; as the frequency and intensity of exercise interventions taken in the 27 papers included in this study varied between studies, as well as the duration of the experiments conducted, these differences may affect the accuracy of the meta-analysis results; finally, this review was not registered for systematic evaluation, and it is recommended to use this in future reviews to better avoid certain scientific biases.

## 6. Conclusions

Thus, taken together, the results of this study demonstrated that aerobic exercise could be an effective adjunct to drug addiction treatment, and be beneficial in improving health during and after detoxification. This systematic review and meta-analysis assessed the effectiveness of aerobic exercise intervention in drug addicts, including physical fitness, emotional state and mental health. Some of the studies included in this study have varying degrees of heterogeneity and there is still insufficient evidence to suggest a clinical advantage in improving mass and BMI in drug addicts. Therefore, a multicenter, large-sample RCT and more high-quality, homogeneous studies to further validate the results of this study are recommended for future studies, thus providing a more reliable evidence-based basis for clinical practice.

## Figures and Tables

**Figure 1 ijerph-20-02272-f001:**
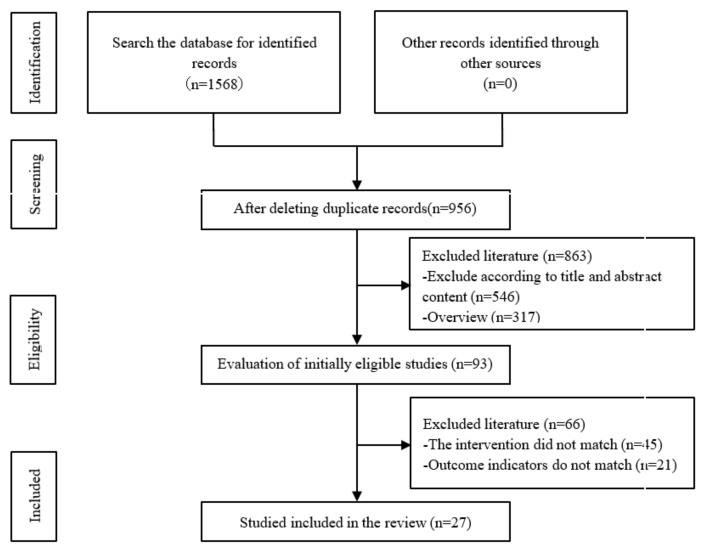
Research screening and selection process according to PRISMA.

**Figure 2 ijerph-20-02272-f002:**
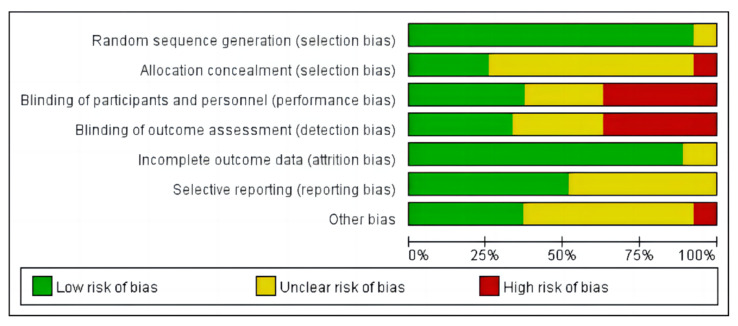
The risk of bias in the included studies.

**Figure 3 ijerph-20-02272-f003:**
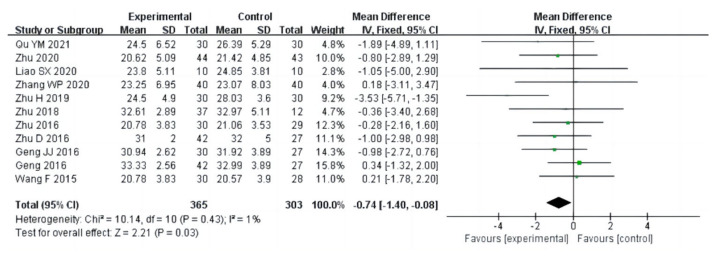
Forest plot comparing body fat percentage in aerobic exercise groups and control groups [12,16,18,19,20,21,25,26,27,28,30].

**Figure 4 ijerph-20-02272-f004:**
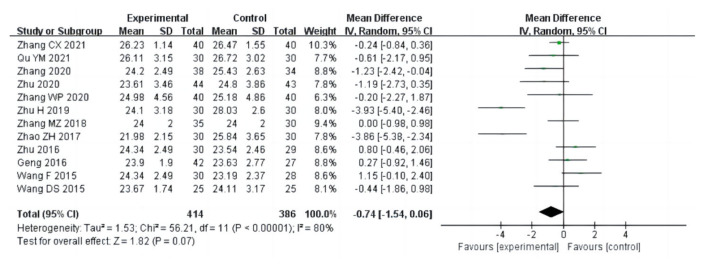
Forest plot comparing BMI of aerobic exercise groups and control groups [12,15,16,17,19,20,22,24,25,26,30,31].

**Figure 5 ijerph-20-02272-f005:**
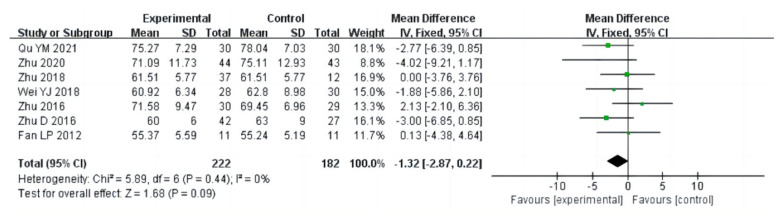
Forest plot comparing mass of aerobic exercise groups and control groups [12,16,21,23,25,27,37].

**Figure 6 ijerph-20-02272-f006:**
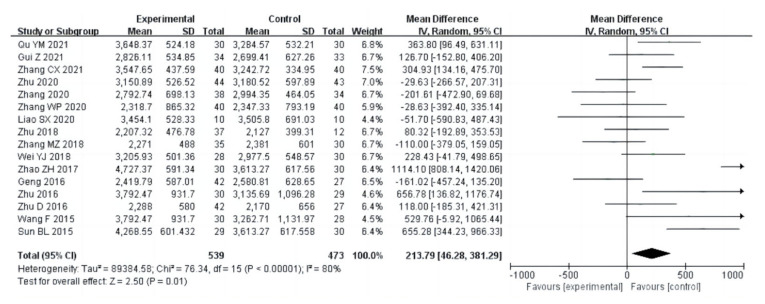
Forest plot comparing vital capacity of aerobic exercise groups and control groups [12,13,15,16,17,18,19,21,22,23,24,25,26,27,30,31].

**Figure 7 ijerph-20-02272-f007:**
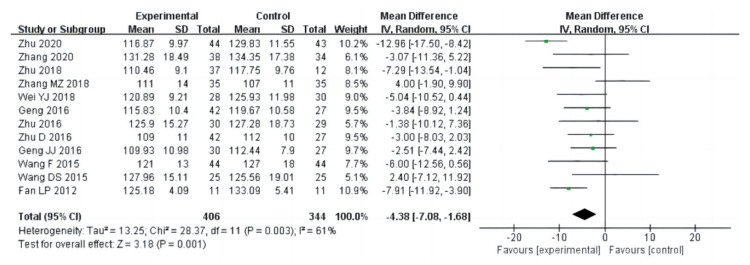
Forest plot comparing systolic blood pressure in aerobic exercise groups and control groups [16,17,21,22,23,25,26,27,28,30,31,37].

**Figure 8 ijerph-20-02272-f008:**
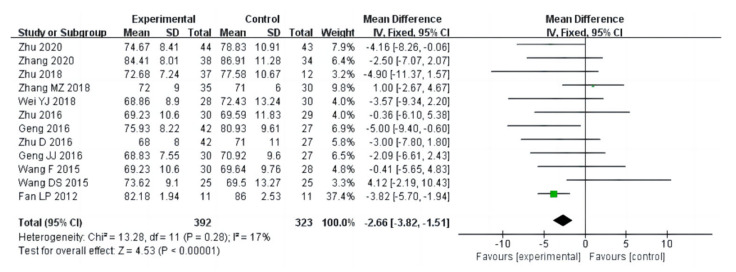
Forest plot comparing diastolic blood pressure in aerobic exercise groups and control groups [16,17,21,22,23,25,26,27,28,30,31,37].

**Figure 9 ijerph-20-02272-f009:**
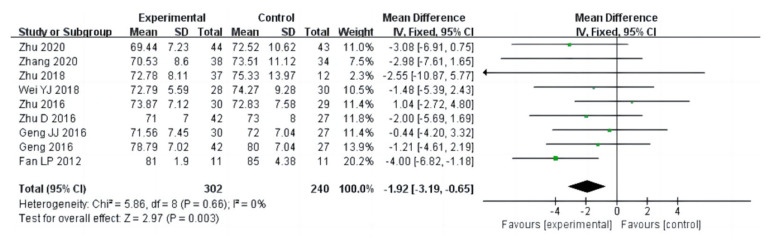
Forest plot comparing beats per minute in aerobic exercise groups and control groups [16,17,21,23,25,26,27,28,37].

**Figure 10 ijerph-20-02272-f010:**
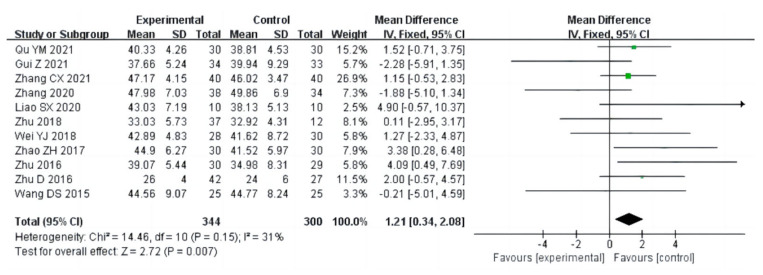
Forest plot comparing muscle force of aerobic exercise groups and control groups [12,13,15,17,18,21,23,24,25,27,31].

**Figure 11 ijerph-20-02272-f011:**
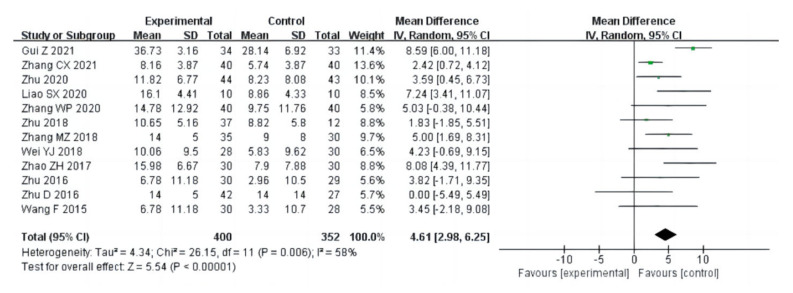
Forest plot comparing flexibility of aerobic exercise groups and control groups [13,15,16,18,19,21,22,23,24,25,27,30].

**Figure 12 ijerph-20-02272-f012:**
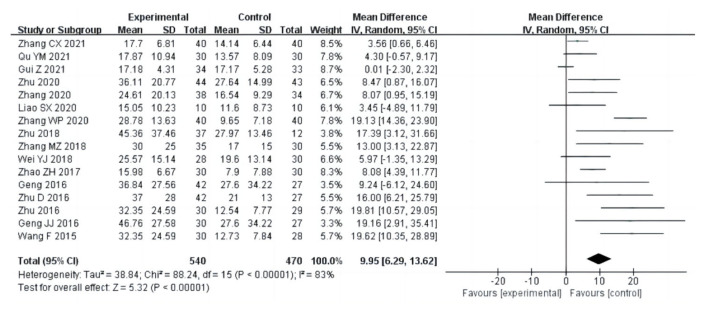
Forest plot comparing the balance in aerobic exercise groups and control groups [12,13,15,16,17,18,19,21,22,23,24,25,26,27,28,30].

**Figure 13 ijerph-20-02272-f013:**
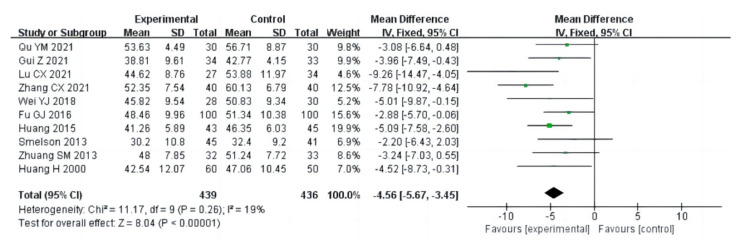
Forest plot comparing anxiety in aerobic exercise groups and control groups [12,13,14,15,23,29,33,35,36,38].

**Figure 14 ijerph-20-02272-f014:**
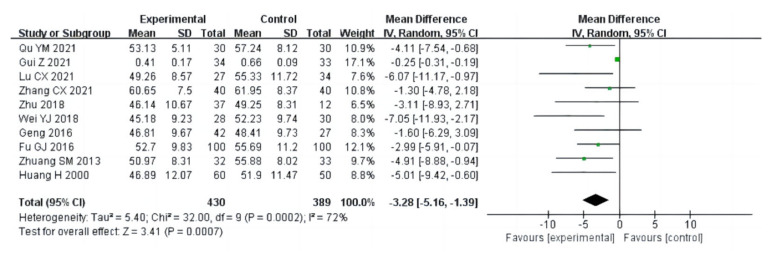
Forest plot comparing depression in aerobic exercise groups and control groups [12,13,14,15,21,23,26,29,36,38].

**Figure 15 ijerph-20-02272-f015:**
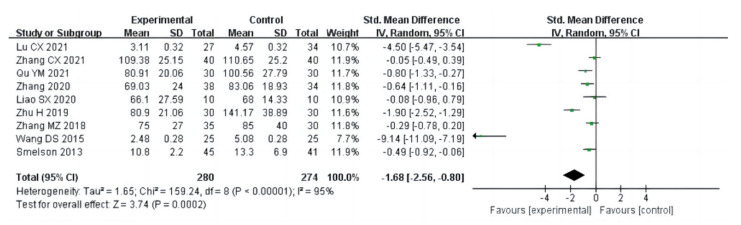
Forest plot comparing drug cravings in aerobic exercise groups and control groups [12,14,15,17,18,20,22,31,35].

**Figure 16 ijerph-20-02272-f016:**
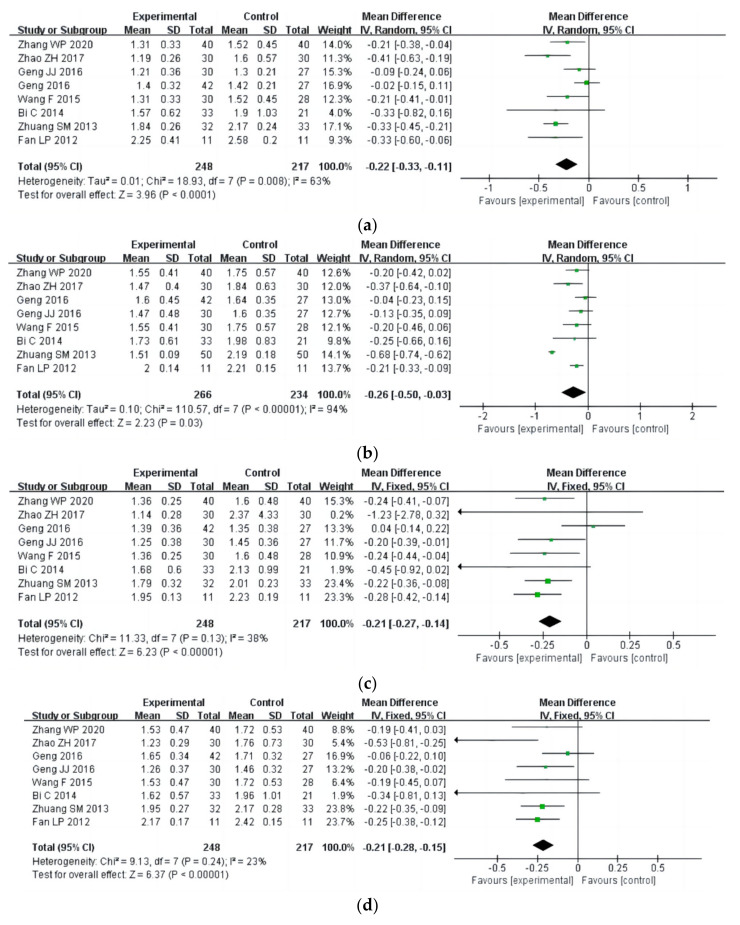
Forest plot comparing the efficacy of each SCL-90 in aerobic exercise groups and control groups [18,23,25,27,29,31,33,35,36]. Note: the figures were focused on the following indicators: (**a**) anxiety; (**b**) obsessive- compulsiveness; (**c**) somatization; (**d**) depression; (**e**) psychoticism; (**f**) phobic anxiety; (**g**) paranoid ideation; (**h**) interpersonal sensitivity; (**i**) hostility.

**Table 1 ijerph-20-02272-t001:** The basic characteristics of the included studies.

Author	Year	Sample	Age(T/C)	Exercise Prescription Variables	Control Group	Outcome
Intensity	Duration(Weeks)	Frequency(Times/Week)	Time(min)
Qu, Y.M. [12]	2021	60	/	High intensity: exercise heart rate should be greater than 75% of the maximum heart rate (HRmax).	24	3	60	Maintaining the original drug treatment for addiction	(1), (2), (3), (4), (8), (10), (11), (12), (13)
Gui, Z. [13]	2021	67	64.23 ± 3.15/64.42 ± 2.91	Moderate	24	5	120	Routine rehabilitation health education and treatment for drug addiction	(4), (8), (9), (10), (11), (12)
Lu, C.X. [14]	2021	61	31.47 ± 0.61/32.32 ± 0.6	/	12	5	/	Routine rehabilitation	(11), (12), (13)
Zhang, C.X. [15]	2021	80	/	Moderate	12	4	60	Maintain original lifestyle habits	(1), (4), (8), (9),(10), (11), (12), (13)
Zhu [16]	2020	100	32 ± 5/30 ± 5	Moderate	12	5	60	Conventional rehabilitation	(1), (2), (3), (4), (5), (6), (7), (9) (10)
Zhang [17]	2020	72	41.08 ± 9.94/39.11 ± 8.9	/	24	5	80	Routine rehabilitation	(1), (4), (5), (6), (7), (8), (10), (13)
Liao, S.X. [18]	2020	20	29.1 ± 3.17/25.6 ± 2.95	65%~75% of HRmax	32	3	60	Routine rehabilitation	(2), (4), (8), (9), (10), (13)
Zhang, W.P. [19]	2020	80	34.56 ± 9.35/35.87 ± 10.45	50%~60% HRmax	12	5	60	Traditional rehabilitation	(1), (2), (4), (9), (10), (14), (15), (16), (17), (18), (19), (20), (21), (22)
Zhu, H. [20]	2019	60	31.73 ± 4.7/32.1 ± 4.99	Moderate	32	4	90	Routine rehabilitation	(1), (2), (13)
Zhu [21]	2018	80	33.74 ± 7.11/37.76 ± 9.85	Moderate	24	3–5	60	Standard care	(2), (3), (4), (5), (6), (7), (8), (9), (10), (12)
Zhang, M.Z. [22]	2018	65	34.8 ± 8.14/34.7 ± 8.39	Moderate	24	5	20	Routine rehabilitation	(1), (4), (5), (6), (9), (10), (13)
Wei, Y.J. [23]	2018	58	30.18 ± 3.7/29.39 ± 3.4	Low to medium exercise intensity	17	3	60	Routine rehabilitation	(3), (4), (5), (6), (7), (8), (9), (10), (11), (12)
Zhao, Z.H. [24]	2017	60	T + C:29 ± 5.28	60%~79% of HRmax at moderate intensity	12	3	60	Daily rehabilitation activities	(1), (4), (8), (9), (10), (14), (15), (16), (17), (18), (19), (20), (21), (22)
Zhu [25]	2016	59	37.47 ± 8.41/42.69 ± 11.37	Moderate	24	3~5	60	Standard care	(1), (2), (3), (4), (5), (6), (7), (8), (9), (10)
Geng [26]	2016	82	34 ± 7/38 ± 10	4.5 MET	24	3~5	60	Traditional rehabilitation	(1), (2), (4), (5), (6), (7), (10), (12), (14), (15), (16), (17), (18), (19), (20), (21), (22)
Zhu, D. [27]	2016	82	34 ± 7/38 ± 10	4.5 MET	12	3–5	60	Routine rehabilitation	(2), (3), (4), (5), (6), (7), (8), (9), (10)
Geng, J.J. [28]	2016	60	34 ± 7.11/37.74 ± 9.88	/	12	5	45	Routine rehabilitation	(2), (5), (6), (7), (9), (10), (14), (15), (16), (17), (18), (19), (20), (21), (22)
Fu, G.J. [29]	2016	200	28.3 ± 7.83/27.99 ± 8.17	Moderate	20	7	30	Routine rehabilitation	(11), (12)
Wang, F. [30]	2015	60	37.47 ± 8.4 41.57 ± 10.53	40%~60% target heart rate	12	5	45	Traditional rehabilitation	(1), (2), (4), (5), (6), (9), (10), (14), (15), (16), (17), (18), (19), (20), (21), (22)
Wang, D.S. [31]	2015	50	32.2 ± 6.97 34.76 ± 7.96	Moderate intensity 65~75% HRmax	12	3	30~40	Routine rehabilitation	(1), (5), (6), (8), (13)
Sun, B.L. [32]	2015	60	29 ± 5.28	/	12	3~5	60	Daily rehabilitation activities	(4), (18), (19), (20)
Huang [33]	2015	100	35.26 ± 12.22/35.21 ± 12.12	/	20	/	30	Psychological therapy	(11)
Bi, C. [34]	2014	95	/	Moderate	12	3	60	Daily routine	(14), (15), (16), (17), (18), (19), (20), (21), (22)
Smelson [35]	2013	101	36.0 ± 9.4/40.4 ± 11.9	/	2	4~6	15	Sham external Qi-therapy (EQT)	(11), (13), (18)
Zhuang, S.M. [36]	2013	65	27.41 ± 5.99/29.06 ± 5.4	55~69% HRmax	12	5	40~50	No exercise intervention	(11), (12), (14), (15), (16), (17), (18), (19), (20), (21), (22)
Fan, L.P. [37]	2012	22	29.73 ± 1.98/27.64 ± 1.69	/	8	5	60	No exercise intervention	(3), (5), (6), (7), (14), (15), (16), (17), (18), (19), (20), (21), (22)
Huang, H. [38]	2000	110	27. ± 1.8/28.2 ± 2.1	50~60% 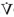 O2max	24	3	40~60	Daily life activities	(11), (12)

Note: T, aerobic exercise group; C, control group; (1), BMI; (2), body fat percentage; (3), mass; (4), vital capacity; (5), systolic blood pressure; (6), diastolic blood pressure; (7), beats per minute; (8), grip strength; (9), sit-and-reach; (10), one-leg stand with eye closed; (11), SAS; (12), SDS; (13), drug craving; (14~22), SCL-90.

## Data Availability

Not applicable.

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
