# Peer review of "Intervention Effect of Aerobic Exercise on Physical Fitness, Emotional State and Mental Health of Drug Addicts: A Systematic Review and Meta-Analysis"

_ijerph, 2023, doi:10.3390/ijerph20032272_

Round 1

Reviewer 1 Report

Two suggestions:

1. Table 1 contains a lot of data, which makes them difficult to read. If it can be split, please do so.

2. Please create a paragraph for the limitations.

Reviewer 2 Report

A good work.

Congratulations

Reviewer 3 Report

First of all, congratulations for the research work done, then I will mention a single recommendation with the aim of obtaining a review that avoids certain scientific biases. 

The main review guidelines recommend the registration of the search before conducting the same for this there are different platforms such as PROSPERO. I ask that if you have made such a record indicate it in the text and if not, I recommend its use in future reviews. Furthermore, they should clearly include this lack in the limitations section of the study.

I hope this observation can be useful and help you to further strengthen this manuscript.

Reviewer 4 Report

1. This study focuses on drug (substance) usage. However, the “drug” was not defined clearly in the article. Although it is obvious that the authors are discussing illegal substances, “drugs” are also commonly used to describe regular medicine pills, or even dietary supplements in some cultures. I suggest that the authors define the “drug,” preferably clearly identifying the name of the substances researched, in the introduction session.  

2. The “endurance exercise” should be defined. What exercises are considered endurance exercises during data selection and extraction? I suggest that the authors define the endurance exercise, preferably listing all the exercises (or exercise categories) used in the data selection and extraction process.  

3. In the data selection and data extraction session, the authors stated that the selection of data was performed by two researchers reading the abstracts and then the contents when deemed appropriate. However, this selection process may raise concerns regarding selection bias. How did the authors ensure that the studies chosen did not favor the authors’ interests? For example, the researchers may deliberately exclude or unfavor the studies that showed no beneficial effects of exercise on drug addiction prevention. The authors should explain how selection bias was prevented during the data selection process. 

4. Line 358-359: The authors need to provide the literature or reference that supports their statement “the phenomenon of anxiety and depression in drug addicts mainly comes from ambivalence about drugs.” 
